# Out of Sight: A Framework for Egocentric Active Speaker Detection

## Abstract

Current methods for Active Speaker Detection (ASD) have achieved remarkable performance in commercial movies and social media videos. However, the recent release of the Ego4D dataset has shown the limitations of contemporary ASD methods when applied in the egocentric domain. In addition to the inherent challenges of egocentric data, egocentric video brings a novel prediction target to the ASD task, namely the camera wearer's speech activity. We propose a comprehensive approach to ASD in the egocentric domain that can model all the prediction targets (visible speakers, camera wearer, and global speech activity). Moreover, our proposal is fully instantiated inside a multimodal transformer module, thereby allowing it to operate in an end-to-end fashion over diverse modality encoders. Through extensive experimentation, we show that this flexible attention mechanism allows us to correctly model and estimate the speech activity of all the visible and unseen persons in a scene. Our proposal named "Out of Sight" achieves state-of-the-art performance in the challenging Ego4D Dataset, outperforming previous state-of-the-art by at last 4.41%.

## 1 Introduction

Active speaker detection (ASD) is a multimodal video understanding task with real-world applications, for example, video conferencing, movie summarization, and video editing. The main goal of ASD is to identify which person (if any) is speaking in an arbitrary video scene, where multiple people could be visible at any moment Roth et al. (2020). The interest of the research community has driven significant advances in the ASD task Alcázar et al. (2020); Chung (2019); Roth et al. (2020); Tao et al. (2021), enabling effective methodologies that approach the ASD task in commercial videos Alcázar et al. (2022); Köpüklü et al. (2021); Tao et al. (2021) and social media clips Alcázar et al. (2021). Despite these current advances in ASD and the ample corpus of research on egocentric data Bambach et al. (2015); Damen et al. (2018); Furnari & Farinella (2019); Huang et al. (2016); Kazakos et al. (2019); Li et al. (2013; 2015); Lu & Grauman (2013), the ASD task in an egocentric setup remains largely under-explored, mostly due to the absence of a standard benchmark for audiovisual detection in the egocentric domain.

The release of the Ego4D dataset Grauman et al. (2022) established the first test-bed to assess the performance of ASD methods in a large-scale collection of egocentric data. Approaching the ASD task in the egocentric domain brought novel challenges to it, among them: fast and continuous camera motion caused by head-mounted capture devices, blurred and smaller visual targets, partial face detections, and an overall less controlled recording environment Grauman et al. (2022); Jiang et al. (2022). We visualize some of these challenging conditions in Fig. 1.

Remarkably, the nature of egocentric data introduced a new evaluation target. In addition to detecting the speech activity of the visible persons on the scene, egocentric ASD also allows for the detection of speech events generated by the camera wearer. Since egocentric video is captured with a head-mounted device, the camera wearer's face is always outside the camera's Field of View, thus making him/her an *off-screen speaker* with no visual patterns. Such an evaluation target remains under-explored in the ASD literature, as current methods focus on attributing speech events to visual targets while avoiding learning from off-screen speech (i.e. off-screen speech is considered a source of noise Alcázar et al. (2021); Roth et al. (2020)).

Commercial Movies | Egocentric Scenes

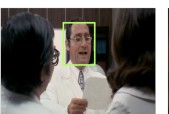 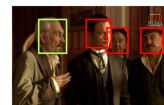 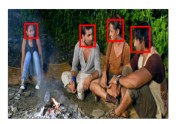 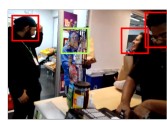 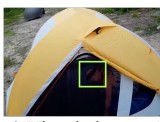 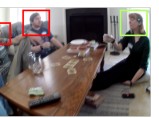

a) Frontal faces    b) No clutter    c) Slow camera motion    d) Non-frontal faces    e) Visual clutter    f) Fast camera motion

Figure 1: **Egocentric Active Speaker Detection.** Active Speaker Detection in commercial movies typically deals with less cluttered scenes **a)** (movie sets), relatively large and frontal faces **b)**, and controlled camera motion. In the egocentric domain fast camera, motion, and blurry visual targets **f)** are far more common. Moreover, the camera wearer (who always remains unseen due to the head-mounted capture device) will also be a valid target although his visual patterns are not available. We propose a sequence-to-sequence learning strategy that overcomes many of these limitations and provides a unified strategy to predict the speech activity of all the targets (visible or not) in the egocentric active speaker detection task.

Under these circumstances, the accurate detection and assignation of speech events would require a major shift from the established audio-to-visual feature alignment paradigm Chung (2019); Köpüklü et al. (2021); Roth et al. (2020); Zhang et al. (2019), to a more flexible multimodal modeling strategy, where some audio events must be modeled without establishing a direct cross-modal correspondence. In this paper, we depart from the standard audiovisual assignment strategy that dominates exocentric ASD, and approach the egocentric ASD task from a novel perspective. Our proposal aggregates all the relevant input data in a video clip regardless of its modality (*i.e.* face crops and audio stream) as a feature sequence, and learns a multimodal embedding that encodes the speech activity of every element. Critically, our model estimates a hidden feature sub-set that enables the disentanglement of speech events into seen and unseen sources, allowing us to perform a fine grain labeling of speech activities into camera wearer events or active speakers utterances.

We draw insights from recent advances in generative auto-encoders Devlin et al. (2018); Feichtenhofer et al. (2022); Fu et al. (2021); He et al. (2022); Ma et al. (2022); Wei et al. (2022), and show that the sequence embedding and reconstruction can be simultaneously modeled in a single transformer module endowed with learnable tokens. To this end, we deploy a transformer architecture where the encoder specializes on multimodal feature fusion, and the decoder specializes on dense prediction and token reconstruction. We name our method Out of Sight ($O^2S$) a sequence-to-sequence transformer that simultaneously models all of the 3 main prediction targets in egocentric audiovisual localization, namely: Voice Activity Detection (VAD), Active Speaker Detection of visible targets (vASD), and egocentric Active Speaker Detection (eASD).

Our work brings the following contributions. (i) We propose a novel framework for egocentric video where we simultaneously model all the 3 key prediction tasks (VAD, vASD, eASD). We show that concurrent modeling increases the performance of all these prediction targets (ii) We show that our proposal can be fully instantiated inside a single transformer architecture and provide state-of-the-art results in the challenging Ego4D Benchmark.

## 2 RELATED WORK

Multiple research efforts have been dedicated to the study of the Active Speaker Detection task. From classical approaches that perform audiovisual feature fusion with time-delayed neural networks Waibel et al. (1989), into strategies that limit the analysis to visual patterns Saenko et al. (2005) only, to more recent works Chakravarty et al. (2016); Chung & Zisserman (2016) which started addressing the more general (and more challenging) multi-speaker scenario.

The recent explosion of deep neural architectures Hara et al. (2018); He et al. (2016); Krizhevsky et al. (2012); Vaswani et al. (2017) has driven the reformulation of the ASD task into the domain of multi-modal representation learning. This reformulation resulted in state-of-the-art methods whose main focus is the design and fusion of deep multi-modal embeddings Chung et al. (2018); Chung & Zisserman (2016); Nagrani et al. (2017); Tao & Busso (2017). Currently, the ASD detection parading is dominated by deep modality-specific models (3D CNNs and transformers) which rely on

modeling a compatible representation space for the involved modalities (audio and visual) Alcázar et al. (2022); Köpüklü et al. (2021); Zhang et al. (2021). These deep networks (designed for the image domain) are used even to process the 1D audio signal Roth et al. (2020).

**Active Speaker Detection in Commercial Movies**   The release of the first large-scale test-bed for the ASD task Roth et al. (2020) has driven the use of deep convolutional encoders Köpüklü et al. (2021) for the ASD task. In addition, the availability of large-scale data has enabled current approaches to shift their focus from maximizing the correlation between a single visual target and the speech event Chung & Zisserman (2016); Roth et al. (2020); Zhang et al. (typically by optimizing a Siamese Network with modality-specific streams), into a more suitable modeling paradigm, where a set of visual targets is jointly modeled along with the audio signal Alcázar et al. (2020); Zhang et al.. Such a process generates an effective assignment, where the visual embedding with the higher affinity is assigned to the speech event Alcázar et al. (2021).

**Multi-Speaker Modeling Mechanism**   One of the earliest multi-speaker strategies was presented in *et al.* Alcázar et al. (2020). This work proposed a feature stack that represents all the active speakers in the scene, the features in this data structure share some temporal overlap and are jointly modeled by using non-local convolutions Vaswani et al. (2017); Wang et al. (2018) and LSTM networks Hochreiter & Schmidhuber (1997). In a similar spirit, follow-up works have modeled this contextual information with 1D CNNs Zhang et al. (2021), a stack of LSTM layers Köpüklü et al. (2021); Zhang et al. (2021), graph neural networks Alcázar et al. (2021); Min et al. (2022), and recently multi-modal transformers Tao et al. (2021).

**Active Speaker Detection in Egocentric Data**   Recently, the interest of the community has shifted from commercial movies into an egocentric setup Jiang et al. (2022), this shift has been driven mainly by the release of the large-scale egocentric dataset Ego4D Grauman et al. (2022), which contains an audiovisual split that allows the study of multiple audiovisual tasks, among them egocentric Active Speaker Detection (ASD). Unlike commercial movies, egocentric data has the unique property of camera wearer speech, for such events no visual target is available, and the common principle of audio-to-visual assignment is broken.

In this paper, we depart from the standard audio-to-visual approach and aim for a more flexible approach to audiovisual localization in the egocentric domain. We rely on flexible contextual modeling where we model the temporal nature of the video and allow every element audio and visual to attend to each other. This simple attention mechanism allows us to simultaneously model the cross-modal correspondences and the single-modality predictions into a single neural architecture.

## 3   OUT OF SIGHT

We begin this section with a general formulation for the egocentric active speaker detection (ASD) problem, then we delve into the details of our proposed approach. Overall, we design an encoder-decoder architecture whose input is a multi-modal feature representation of a video clip. This feature sequence is first encoded to allow information sharing among the modalities (video and audio). The sequence is later decoded into another set that contains predictions for each of the targets involved in egocentric ASD (VAD, vASD, and eASD). For simplicity, we first describe how our approach operates over a short video snippet, then we extend the analysis to multiple temporally adjacent clips. Figure 2 shows an overview of our approach operating over a single video snippet.

### 3.1   PROBLEM FORMULATION AND NOTATION

Active speaker detection (ASD) in egocentric video identifies whether or not there is speech activity at a given moment. In addition, it must discern if any of the visible persons or the camera wearer (who is always outside the Field of View) is the source of the speech event.

For any given frame there are $k \geq 0$ visible faces denoted as $X = \{x_1, x_2, \ldots, x_k\}$, these facial crops constitute the visual modality on the egocentric ASD problem. We complement this visual input with the associated audio stream ($s$), and a learnable feature set ($c$) which represents the unseen camera wearer speech target. We define the input set for the egocentric ASD task as $\{X, s, c\}$

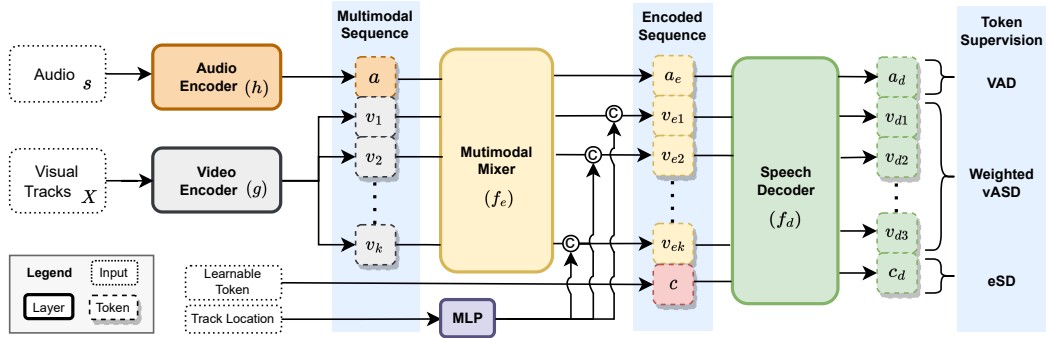

Figure 2: **Out of Sight Architecture, Short-Term.** Out of Sight relies on two modality encoders (video encoder in light gray and audio encoder in orange). These encoders produce a feature set that is arranged as a sequence (labeled "multi-modal sequence"). This sequence is then encoded allowing for the information sharing between the two modalities. Upon encoding, it is extended with a learnable token $c$ (in light red). We estimate another sequence-to-sequence mapping with the speech decoder (light green) which allows us to generate a prediction for each of the mentioned tokens. Out of Sight models 3 prediction targets simultaneously: Voice Activity Detection (obtained from the $a_d$ token), egocentric Active Speaker Detection (obtained from the $v_{di}$ tokens), and camera wearer speech detection (from the $c_d$ token). To this end, we supervise each task independently (see the "Token Supervision" heading).

and the corresponding prediction set as $Y = f(X, s, c) = f(g(x_1), \ldots, g(x_k), h(s), c)$. We note that $Y$ is also a sequence that contains confidence predictions for each of its elements, that is: $Y = \{y_{x1}, \ldots, y_{xk}, y_s, y_c\}$. The sub-sequence $\{y_{x1}, \ldots, y_{xk}\}$ contains the predictions for the visual targets (vASD), $Y_s$ the prediction for the global speech activity (VAD), and $Y_c$ the prediction for the camera wearer speech activity (eASD).

## 3.2 OUT OF SIGHT ARCHITECTURE

Our proposed architecture consists of 3 main building blocks: (i) Modality Encoders (audio/video), (ii) Multimodal Mixer, and (iii) Speech Decoder. The modality encoders operate over the face crops $x_k$ and audio stream $s$. Following the standard practice in the literature Chung (2019); Köpüklü et al. (2021); Roth et al. (2020); Tao et al. (2021); Grauman et al. (2022), our modality encoders are two independent convolutional backbones: a 3D or hybrid (3D and 2D) encoder for the visual stream $g(x_i) = v_i$ and a 2D encoder for the audio stream $h(s) = a$. Independent of the modality, the feature embedding estimated on each forward pass is assigned to an individual token for a total of $k + 1$ tokens ($k$ from the visual stream, 1 from the shared audio track).

The Multimodal Mixer ($f_e$) is instantiated as a transformer encoder. It takes as an input the initial set of $k + 1$ multimodal tokens $\{a, v_1, \ldots v_k\}$, and serves one main purpose: it performs cross-modal attention on the multimodal sequence thus aggregating relevant information across the two modalities. As a consequence, $f_e$ estimates an initial sequence-to-sequence mapping $\{a, v_1, \ldots v_k\} \rightarrow \{a_e, v_{e1}, \ldots v_{ek}\}$, we name this intermediate set the *encoded sequence*. We show the multimodal mixer in light yellow in Figure 2).

The Speech Decoder module ($f_d$) is instantiated as a transformer decoder, it performs two main tasks. First, it maps the tokens contained in the encoded sequence into a *decoded sequence* that directly models the speech activity of each individual element. Second, it estimates a suitable feature set for the camera wearer from a placeholder token. Considering that there is no visual information for the camera wearer, and the audio features are already encoded in the token $a_e$, we inject a learnable token ($c$) in the encoded sequence. The attention mechanism in the speech decoding refines the feature representation of this place-holder token into a suitable embedding that allows us to estimate the speech activity of the camera wearer. We show the multimodal mixer in light green in Figure 2, and the learnable token $c$ is depicted in light red in the same figure.

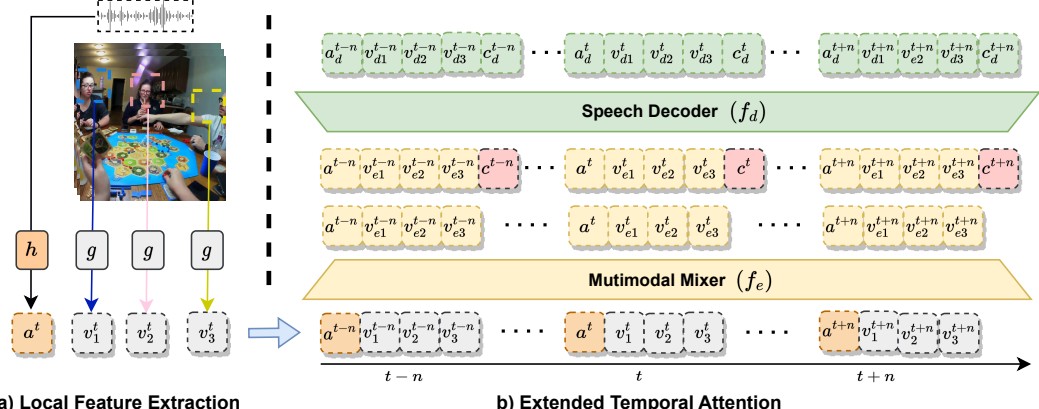

a) Local Feature Extraction          b) Extended Temporal Attention

Figure 3: **Out of Sight Architecture, Long-Term.** At the time $t$ we sample up to $k$ face tracks (in this figure $k = 3$) and the associated audio stream as the input to the Out of Sight (subplot a). We estimate the feature set for every face crop $v_j^t, j \in [1, k]$ and the associated features for the audio spectrogram $a^t$. We extend this local analysis $n$ times forward and backward and assemble all the individual features into a single sequence of length $(2n + 1)(k + 1)$, we simply concatenate and forward pass this entire sequence on the encoder and decoder. Therefore, long-term modeling enables attention to any element in the extended temporal sequence.

The decoder $f_d$ learns another sequence-to-sequence map. Since the camera wearer token is appended into the encoded sequence, the output has a total of $k + 2$ elements *i.e.* $\{a_e, v_{e1}, \dots v_{ek}\} \rightarrow \{a_d, v_{d1}, \dots v_{dk}, c_d\}$. The speech decoder maps all the tokens (audio, visible faces, and camera wearer) into a joint representation space. Therefore, we can obtain the final prediction for any element in the decoded sequence using the same linear layer. The supervision scheme is outlined in Figure 2, see the heading "Token Supervision".

**Short-term Feature Modeling.** At the time $t$, we estimate the associated mel-spectrogram of the audio signal centered at $t$. For every visible person, we sample a short clip of temporally contiguous face crops centered at the same timestamp, thus building $k$ tracks centered at time $t$. Our modality encoders $(g, h)$ generate the initial feature tokens from the spectrogram and the tracks (potential speakers) sampled around time $t$, see Figure 3 a).

**Long-term Feature Modeling.** Since we rely on 3D or hybrid feature extractors, each feature embedding $v_k$ already encodes short temporal patterns extracted from the stack of time-contiguous face crops $x_k$. However most ASD methods Alcázar et al. (2020); Köpüklü et al. (2021); Tao et al. (2021) rely on longer temporal sampling windows to achieve improved performance. Therefore, we incorporate an extended temporal sampling similar to Alcázar et al. (2021) and replicate our local analysis along multiple timestamps centered at time $t$ (see Figure 3 b)). We define $n$ additional sampling points ahead and behind time $t$ obtaining a total of $2n + 1$ temporal samples $(t - n, \dots, t, \dots, t + n)$. Unlike Alcázar et al. (2021), this extended sampling does not imply any additional message-passing structure or network extensions. It simply increases the length of the input sequence. The multimodal and encoded sequences now have $(2n + 1)(k + 1)$ elements, and the decoded sequence now has a total length of $(2n + 1)(k + 2)$ tokens. The rest of our approach remains unaltered.

### 3.3 LEARNING FROM EGOCENTRIC DATA

In addition to our encoder-decoder architecture, we include 2 domain-specific adaptations to further enhance the performance of ASD in egocentric videos.

**Visual Token Representation.** We augment the feature representation of the visual tokens $v_k$ with the relative position of the face crop in the scene. Previous works Zhang et al. (2021) have shown

this prior information improves ASD performance in commercial movies. In the egocentric domain, this modification could correlate with the gaze direction, as humans often gaze at the person who is speaking. In other words, the active speaker is more likely to appear near the center of the frame.

The speaker's relative position is represented by the normalized corner coordinate of the face crops; we forward this 4 position vector along 2 MLP layers (see Figure 2 the purple box) and append it to the feature of the visual tokens $v_k$ just before the speech decoder.

**Weighted Visual Loss.** We observe a large number of noisy face detections, including non-frontal face crops, blurred faces, and partial detections which are not useful for ASD. We propose to mitigate the undesired noise from such faces, by adding a data-dependent weight factor to the loss function. We lower the contribution of individual face detections according to their noise level, and set a threshold for extremely noisy detections so they are disregarded in the loss calculation.

We approximate the noise level in each face by the confidence score obtained from a generic face detector. We use the single-stage detector of Deng et al. (2020), and obtain the detection confidence for each individual face crop on the dataset. We avoid propagating the losses of the visual tokens if their score is below a fixed threshold $\alpha$. In every other case, we weight the individual loss contribution of every $v_{di}$ by the detection score of the corresponding $x_i$. Since we only need the final face detection score of every face crop, we pre-compute these values before training our method. Therefore, Out of Sight is still end-to-end trainable, and has no additional parameters or added FLOPS due to the face detector.

**Token Supervision.** At training time we independently supervise every token in the decoded sequence using cross-entropy loss. The token $a_d$ is supervised with the global speech activity ($\mathcal{L}_a$). The Token $c_d$ is supervised with the camera wearer's speech activity ($\mathcal{L}_c$). And each of the $v_{dk}$ tokens are supervised with the corresponding active speaker ground-truth ($\mathcal{L}_v$) weighted by the corresponding face detection score ($d$), and $\alpha$ threshold ($\mathbb{1}_{d \geq \alpha}(d)$). Formally:

$$\mathcal{L} = \alpha \mathcal{L}_a + \mathcal{L}_v [\mathbb{1}_{d \geq \alpha}(d)] + \beta \mathcal{L}_c \tag{1}$$

Where $\alpha$ and $\beta$ are weight factors incorporated into our proposal as the audio stream exhibits much faster convergence Wang et al. (2020). The decoded sequence already contains all the relevant predictions for the VAD, egoASD, and egoVAD tasks. Under long-term sampling, we follow the outlined supervision strategy for each of the sampling windows.

### 3.4 IMPLEMENTATION DETAILS

We implement Out of Sight using the PyTorch framework Paszke et al. (2019). We use the RAdam optimizer Liu et al. (2019), set the learning rate to $7e^{-4}$ and implement gradient clipping limiting the norm to $1.0$. Regarding the modality encoders, we adopt ResNet-18 He et al. (2016) as the audio encoder and average the weights of the original input layer to account for the single-channel mel-spectrogram. We use the R3D 18-layer as the 3D video encoder Hara et al. (2018). For the hybrid video encoder, we inflate the initial layer of a 2D ResNet-18 Carreira & Zisserman (2017); Chung (2019) and pool along the temporal dimension right after the first layer, thus preserving the original 2D architecture from the second layer onward.

After obtaining the initial feature set from the modality encoders. We reduce the feature dimensions for both audio and video to $128$ using two linear layers (one per modality) and keep this feature size stable throughout the transformer encoder and decoder. The transformer encoder and decoder follow the same architecture and are composed of one self-attention module Vaswani et al. (2017). All the self-attention layers have 8 heads, and their output is post-processed with 2 linear layers including Layer Normalization Ba et al. (2016) which follows the pre-normalization pattern proposed by Shoeybi et al. (2019).

We use binary cross-entropy loss to supervise individual tokens of the decoded sequence and apply the weighting scheme described in 3.3 only for the visual tokens, $\alpha$ is set to 0.5 and $\beta$ is set to 0.5. We train end-to-end on two NVIDIA-V100 GPU using the accelerate library Sylvain Gugger (2022) with mixed precision and a batch size of 26 per GPU. The face crops are resized to a fixed resolution of $160 \times 160$, our largest model (Out of Sight 3D) converges in under 12 hours. Our best results are

| Method | Ego4D Pretrain | AVA Pretrain | Prediction Smoothing | vASD mAP | eASD mAP |
|---|---|---|---|---|---|
| Ego4D - Audio Matching Grauman et al. (2022) | ✓ | ✗ | - | - | 43.95 |
| Ego4D - ResNet-18 Grauman et al. (2022) | ✓ | ✗ | - | - | 72.00 |
| Min *et al.* Min (2022) | ✓ | ✗ | - | - | 80.40 |
| Ego4D - RegCls Grauman et al. (2022) | ✓ | ✗ | ✗ | 20.33 | - |
| Ego4D - RegCls Grauman et al. (2022) | ✓ | ✗ | ✓ | 24.60 | - |
| Ego4D - Talknet Grauman et al. (2022) | ✓ | ✗ | ✓ | 51.04 | - |
| LoCoNet Wang et al. (2023) | ✓ | ✗ | ✗ | 59.69 | - |
| **Out of Sight 2.5D (Ours)** | ✓ | ✗ | ✗ | **63.10** | **83.85** |
| **Out of Sight 3D (Ours)** | ✓ | ✗ | ✗ | **64.10** | **84.81** |

Table 1: **State-of-the-art Comparison on Ego4D Validation Set.** We compare Out of Sight to the state-of-the-art methods from Grauman et al. (2022) in the Ego4D validation set. Out of Sight 3D outperforms by 9.2% the current state-of-the-art for vASD and by 12.08% for eASD. When we train Out of Sight on AVA and directly evaluate on Ego4D, Out of Sight obtains 7.52% improvement on vASD when compared to the baseline of Grauman et al. (2022) (Talknet). We also highlight that Out of Sight is the only approach that can simultaneously generate predictions for vASD and eASD.

obtained by setting $n = 2$ and $k = 3$. Each individual video clip ($x_i$) contains 7 frames. In total, our analysis window spans 35 frames (about 1.17 seconds). The average frame rate in Ego4D is 30fps.

**Visual Input Sampling** We set $k = 3$ but clearly not every video frame will contain 3 face detections, we follow the strategy of Alcázar et al. (2020) and sample with replacement whenever there are less than 3 face detections. Whenever there are more than 3 detections we simply sample 3 random detections.

**End-to-end Training** We follow an approach similar to Alcázar et al. (2022) and perform simultaneous forward passes over $k$ visual targets and the associated audio spectrogram. We assemble the multimodal sequence on-the-fly on pre-allocated GPU-RAM buffers. After going through the Multimodal Mixer module, we append the learnable token $c^t$ into the encoded sequence and proceed with the forward pass through the Speech Decoder. With the pre-calculated face detection scores there are no conditionals or variable sizes in the forward graph.

## 4 EXPERIMENTAL RESULTS

In this section, we provide the empirical evaluation of Out of Sight ($O^2S$). We mainly evaluate Out of Sight on the Ego4D Grauman et al. (2022) dataset. We begin this section with a direct comparison to the state-of-the-art. Then, we ablate our main design decisions to assess their individual contributions.

### 4.1 STATE-OF-THE-ART COMPARISON

Table 1 compares Out of Sight against the state-of-the-art in the Ego4D dataset. For the speech detection of visible speakers (vASD). Using a 3D visual backbone our approach improves the mean average precision (mAP) by up to 12.4% over the Ego4D baseline Grauman et al. (2022), and 4.41% over the proposal of Wang et al. (2023). When using a hybrid (2.5D) visual backbone our proposal achieves an improvement of 12.06% for the active speaker detection over the baseline of Grauman et al. (2022). We note that our 2.5D backbone is a simplified version of the visual backbone used in both Grauman et al. (2022) and Wang et al. (2023). We only incorporate the initial 3D convolutional blocks, whereas Grauman et al. (2022) is enhanced with the video temporal convolutional block of Tao *et al.* Tao et al. (2021), and Wang et al. (2023) is extended with the VTCN module of Lea et al. (2016).

Regarding the camera wearer detection (eASD), $O^2S$ achieves a significant improvement over the baseline of Grauman et al. (2022), outperforming by up to 12.81%. We highlight that the baseline of Grauman et al. (2022) uses the exact same 2D ResNet-18 for the audio stream encoding as Out

of Sight, the only difference being that Grauman et al. (2022) relies on slightly shorter sampling windows (1 second v.s. 1.17 seconds for our method). Meanwhile, Min (2022) relies on the pre-trained model of Team (2021) to remove false positives on their predictions. Without any bells and whistles, Out of Sight achieves a significant improvement on the eASD task. We attribute the improved performance to the proposed transformer module that allows the joint modeling of audiovisual cues, instead of the audio-only paradigm in Grauman et al. (2022) and Min (2022).

## 4.2 ABLATION STUDY AND PERFORMANCE ANALYSIS

We now perform the ablation study of Out of Sight and analyze the key design decisions of our approach: the design of the encoder-decoder transformer and the length of the input sequence. We use our best model (Out of Sight 3D) for all the ablation experiments, we do not use any prediction smoothing.

**Out of Sight Components.** We assess the importance of each component in $O^2S$, namely: speech decoder (3.2), visual feature representation enhanced with face location, weighted visual loss (3.3), learnable token 3.3 to model the camera wearer speech activity, and multi-task learning 3.2. In Table 2 we summarize the effects of removing each individual component. Overall, we observe that the speech decoder provides the biggest empirical contribution, Out of Sight drops its performance by 3.15% without it. Since the decoder contains the modules that learn the token $c$ it becomes impossible for the network to make a prediction on the eASD task. The weighted loss is the second most important design contributing with about 2% mAP. We attribute this improvement to the inherently noisy nature of egocentric video, although the weighted loss implies training with less data, this subset has much higher quality.

| Network | Speech Decoder | Weighted Loss | Face Position | Token Parameter | vASD mAP | eASD mAP |
|---------|:---:|:---:|:---:|:---:|:---:|:---:|
| $O^2S$ 3D | ✗ | ✓ | ✓ | ✓ | 60.95 | - |
| $O^2S$ 3D | ✓ | ✗ | ✓ | ✓ | 62.19 | 84.50 |
| $O^2S$ 3D | ✓ | ✓ | ✗ | ✓ | 63.34 | 84.53 |
| $O^2S$ 3D | ✓ | ✓ | ✓ | ✗ | 63.25 | 84.12 |
| $O^2S$ 3D | ✓ | ✓ | ✓ | ✓ | **64.10** | **84.81** |

Table 2: **Out of Sight Components Ablation.** We observe that the largest performance drop appears when removing the speech encoder (about 4.79%). The face localization information and weighted loss both contribute to the final performance to a smaller extent, with the former contributing about 1.2% mAP and the latter about 0.4%.

The learnable token can be approached as either a zero initialized token or as a network parameter, we empirically find that learning a network parameter improves the performance by 0.9% mAP in comparison to simply appending a zero initialized token, this design decision seems to also have a slight impact in the eASD performance reducing it by 0.7% mAP. Finally appending the face location to the visual feature tokens brings a slight improvement around 0.5% to the overall performance, again this seems to have a minimal influence on the performance of the eASD predictions.

**Multi-Task Head** $O^2S$ models the simultaneous predictions of VAD, vASD and eASD. Although the VAD performance is not included in the Ego4D benchmark, we find an empirical benefit for jointly modeling the VAD task. If we set $\alpha$ to 0 we observe a nearly identical performance on the eASD task (84.60%), but the vASD task loses about 1% mAP reporting (63.19%). In the Ego4D validation set we obtained 91.4 mAP for the VAD task.

**Out of Sight Network Depth.** We analyze the effect of the transformer encoder-decoder depth. Table 3 outlines the effect of different depths (2 to 8 layers) in the transformer encoder-decoder. We observe that the best performance is obtained with the 4-layer setup (2 encoder, 2 decoder). Deeper networks overfit faster and show decreased performance after 4 layers. Meanwhile, shallower encoders underperform in our end-to-end proposal. We also note that the depth of the network is not

as critical for the eASD target as for the vASD target. State-of-the-art eASD results are already obtained after stacking only 1 layer per module (2 total).

| Transformer Layers | 2 | 4 | 6 | 8 |
|---|---|---|---|---|
| Additional Params | 262K (0.5%) | 524K (1.1%) | 786K (1.7%) | 1M (2.3%) |
| vASD | 62.95 | **64.10** | 63.1 | 61.11 |
| eASD | 84.41 | **84.81** | 83.82 | 82.25 |

Table 3: **Number of Encoder Layers.** We ablate the number of self-attention layers in the transformer modules. As we increase the depth to 4 layers we get improved performance in the vASD target. However, the eASD task is less sensitive and already obtains state-of-the-art results after stacking 2 layers. We also report the total additional number of parameters and the relative increase with respect to the modality encoders.

**Input Sequence Length.** As mentioned in section 3.2, the length of the encoded sequence depends on two factors, the number of visual targets $k$ per frame and the length of the temporal window samplings $2n + 1$. The overall length of the sequence is given by $l = (2n + 1)(2 + k)$. Tables 4 and 5 show the effect of varying the number of visual targets per sequence and temporal samplings, respectively.

Table 4 shows that the optimal number of visual targets $k$ per frame is 3 for both vASD and eASD targets. This is a stark contrast to the average of visible persons for the Ego4D dataset (0.74 visual targets per frame), but closer to the average number of persons per clip (4.71). Our hypothesis is that, at training time, the network requires a set of visual targets to learn meaningful vASD predictions, otherwise it can overfit to the more frequent scenario with 1 or no visual targets.

| Visual targets ($k$) | 1 | 2 | 3 | 4 |
|---|---|---|---|---|
| Sequence Length ($l$) | 15 | 20 | 25 | 30 |
| vASD | 60.91 | 62.50 | **64.10** | 60.51 |
| eASD | 83.92 | 84.25 | **84.81** | 84.15 |

Table 4: **Number of Visual Targets.** We explore the effect of different numbers of visual targets ($k$) in our multi-modal sequence. Out of Sight obtains the best performance with 3 visual targets (25 total tokens in the sequence). This number slightly correlates with the average number of individuals in every video clip (about 4.7 in Ego4D).

Table 5 shows that the optimal window length is 5 ($n = 2$), with a total of 35 frames. Shorter sequences can not effectively capture the global information required for the vASD and eASD tasks, but longer sequences may bring more noise into the analysis window. Moreover, these longer sequences are much more costly to train, given the number of individual visual samples and the quadratic nature of the attention operation.

| Temporal Samples | $1(n = 0)$ | $3(n = 1)$ | $5(n = 2)$ | $7(n = 3)$ |
|---|---|---|---|---|
| Total Frames in Sequence | 7 | 21 | 35 | 49 |
| vASD | 60.20 | 62.18 | **64.10** | 63.69 |
| eASD | 78.89 | 83.32 | **84.81** | 84.49 |

Table 5: **Number of Temporal Samples.** We assess different sampling window sizes for our proposed method. We find that the optimal setting includes 35 frames (5 contiguous clips of 7 frames each) which span about 1.1 seconds in the Ego4Ddataset. Longer sequences do not benefit the performance and significantly increase the computational effort.

## 5 CONCLUSION

In this paper, we introduced Out of Sight, a novel method for active speaker detection tailored to the challenging egocentric domain. At the core of Out of Sight, there is a transformer encoder-decoder that allows modeling each individual prediction in egocentric ASD as a token. By carefully designing the token supervision, we achieve an end-to-end trainable network that simultaneously solves the vASD task and the eASD tasks. Out of Sight achieves state-of-the-art performance in the challenging Ego4D benchmark in both audiovisual detection tasks.

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
