# OpenReview forum: "Out of Sight: A Framework for Egocentric Active Speaker Detection"
_ICLR.cc/2024/Conference — Submitted to ICLR 2024_

### Official Review · Reviewer_1nZN · 2023-10-29

**Soundness:** 1 poor
**Presentation:** 2 fair
**Contribution:** 1 poor
**Rating:** 1
**Confidence:** 4

**Summary:**

This paper introduces a transformer-based approach, Out of Sight, that can model all the speaking activities from three prediction targets (visible speakers, camera wearer, and global background speech). The proposed method consists of 3 building blocks: Encoder, Multimodal Mixer, and Decoder. The encoder is a series of convolutional networks that embeds the visual and audio features. The multimodal mixer performs cross-modal attention and aggregates relevant information from audio-visual modalities. The decoder maps all types of tokens (audio, visual, and camera wearer) into a common representation space and predicts the final prediction. To further improve the performance, it uses a technique of long-term feature modeling by incorporating an extended temporal sampling.

**Strengths:**

- The proposed architecture is very simple and easy to understand.
- It can be used to predict all the speaking targets (visible speakers, camera wearer, and global background speech) using a single architecture.

**Weaknesses:**

First of all, it looks like the authors provide inconsistent comparisons.
- In Table 1, the authors report vASD mAP scores on the Ego4D’s validation set. The problem is that they report mAP\@0.5 (which is different from mAP) for previous methods when they report mAP only for their method. When we compute the mAP score, we only use the ground-truth face bounding boxes. However, computing mAP\@0.5 involves comparing IoU between the face bounding-box detections and the ground-truth, therefore mAP\@0.5 is estimated much lower than mAP.
- The authors didn’t report the vASD mAP of Min et al. Min (2022) although they report eASD mAP of it. This kind of partial reporting might make their method look more powerful, but doesn’t seem appropriate.
- Reporting unfair and inconsistent comparisons confuses the readers and the whole computer vision community. I believe the authors should be more accurate in describing their validation scheme and the validation strategy of the Ego4D paper.

Second, the proposed method and the results are not state-of-the-art.
- SPELL (2022) and STHG (2023) achieve 71.3% vASD mAP and 75.7% vASD mAP on Ego4D’s validation set, respectively, which significantly outperform the proposed method. Furthermore, STHG (2023) achieves 85.6% eASD mAP, which also outperforms the proposed method in this paper. Please refer to the challenge reports and recognize them. It is recommended by the Ego4D organizers to properly acknowledge their technical reports.

Moreover, the proposed method has some weaknesses in its form.
- For the weighted visual loss, the user needs to pre-compute a weight factor. The weight factor needs to be fine-tuned for each dataset (because it is data-dependent), which is ineffective and seems ad hoc.
- There are many other hyper-parameters that need to be fine-tuned for each dataset: $\alpha$, $k$, $n$, $\beta$, which makes the overall method complicated and hard to optimize and utilize.

[SPELL (2022)] Intel Labs at Ego4D Challenge 2022: A Better Baseline for Audio-Visual Diarization

[STHG (2023)] STHG: Spatial-Temporal Heterogeneous Graph Learning for Advanced Audio-Visual Diarization

**Questions:**

What are the FLOPS and memory requirements for the proposed method? What is the throughput? Most of the previous state-of-the-art approaches are very efficient in terms of FLOPS and memory, and I wonder if the proposed method is comparable.

**Details Of Ethics Concerns:**

- The authors falsely report the performance by treating mAP and mAP\@0.5 as the same metrics. I am worried this can confuse the readers. In addition, the authors only report the scores of the lower-performing models than their proposed method.
- Table 24 of the Ego4D paper (p.60, Ego4D: Around the World in 3,000 Hours of Egocentric Video) shows that all the methods report mAP\@0.5 (not mAP). In addition, the official evaluation tool that all the challenge participants use computes mAP\@0.5. When we compute the mAP score, we only use the ground-truth face bounding boxes. However, computing mAP\@0.5 involves comparing IoU between the face bounding-box detections and the ground-truth, therefore mAP\@0.5 is estimated much lower than mAP. mAP score of previous approaches already reached over 70%, which outperforms the proposed method in this paper.

---

> ### Author Response · Authors · 2023-11-23
> **Evaluation Metric and Comparison to ArXiv pre-prints**
>
> Since the review bases his/her entire rejection argument in the use of the mAP metric and the comparison against SPELL and  STHG, we refer him/her to the global replies titled 'Comparison against ArXiv Submisions' and 'mAP Metric'

---

### Official Review · Reviewer_2WYw · 2023-10-31

**Soundness:** 3 good
**Presentation:** 2 fair
**Contribution:** 2 fair
**Rating:** 3
**Confidence:** 4

**Summary:**

The authors propose a method for active speaker detection of egocentric videos. Unlike existing works that focus on YouTube or broadcast videos, ASD for egocentric videos brings additional challenges such as head movement and camera wearer's speech. The paper proposes a 3-stage architecture consisting of (1) modality encoders, (2) multimodal mixer, (3) speech decoder. In particular, a learnable token helps to model speech from the camera wearer. Short- and long-term architectures enable effective multi-modal fusion and extended temporal modelling. The authors use additional tricks like weighted visual loss and position of face in order to improve performance. The method is evaluated on the recent Ego4D dataset which contains various egocentric videos including human speech.

**Strengths:**

- The method is well-engineered and achieves a strong performance.
- Techniques such as face position and weighted loss paper is well tailored to this dataset.

**Weaknesses:**

- The performance exceeds existing works, but the gap between the proposed method and LoCoNet is not too significant, given that this paper is specifically tailored to this dataset.
- The methods used such as "face position" and "weighted loss paper" are the sources of most of the improvement compared to LoCoNet, but these tricks might be overfitting to the biases that exist specifically in this dataset. Does the method still generalise to existing non-egocentric ASD datasets, such as AVA-ASD or ASW?
- Most similar literature to this is (Jiang et al., 2022) but there is very little comparison to this work in the paper.
- I am not sure if "active speaker detection" is an appropriate term for the overall task. vASD in this paper is usually called ASD in other literature, and the ASD usually does not encompass what is called eASD in this paper. A similar work (Jiang et al., 2022) does not refer to this task as ASD.
- Regardless of the term used, I am not sure if the proposed combined problem (eASD+vASD) is useful, in between vASD and AV speaker diarisation. The camera wearer is a specific identity, whereas we do not consider the identity information of the visible speakers.

Regarding clarity/writing:
- The authors use abbreviations egoASD/egoVAD in page 6, but these are not explained or used anywhere else.
- What is meant by "visible" and "unseen" exactly? Why not "seen" and "unseen" or "visible" and "invisible" for example?
- Is "at last 4.41%" at the end of abstract is the intended expression?
- "fine grain..." in the introduction should be "fine-grained..."
- Sec 3.2 refers to "Token Supervision" without section number, but this only appears much later making it confusing.

**Questions:**

- Please see questions in 'weaknesses'.
- Does the proposed method work well when there are multiple off-screen speakers? For example, it is realistic to have off-screen speakers next to the camera wearer in a meeting.
- Do the authors use the pre-trained weights for LoCoNet, or is it re-trained on the same dataset?

---

> ### Author Response · Authors · 2023-11-23
> **Performance vs LoCoNet.**
>
> While LoCoNet has a close performance in the vASD task we highlight that Out-of-Sight has a much simpler architecture (we dont model inter-speaker or long-term single speaker relationships), despite the simple architecture, direct and accurate estimates for the egoASD and vASD emerge from the direct multi-modal modeling without using the spatial and temporal attention schemes proposed in LoCoNet. In a single end-to-end trainable model Out-of-Sight can model both the egoASD and vASD.
> For further comparison, LoCoNet requires an input tensor of up to 200 frames to achieve optimal performance, meanwhile O2S achieves improved results covering only 35 frames.

---

> ### Author Response · Authors · 2023-11-23
> **Face position and Weighted loss.**
>
> Certainly O2S learns from  global spatial patterns found in Egocentric video (as shown by the improved performance using the face location). As outlined in the paper this correlates to the camera wearer’s  gaze. However,  using the face positions is not exclusive to O2S or Egocentric video. Zhang et al. have already demonstrated the usefulness of this information source in commercial movies (where being at the center of the scenes correlates with directing the attention of the audience to an important character).
> We see this additional information as common prior in the ASD task, regardless of the setup, egocentric or commercial movies. In the egocentric domain, we are the first to show the usefulness of this spatial prior.

---

> ### Author Response · Authors · 2023-11-23
> **Terminology**
>
> We diverted slightly from the standard terminology trying to emphasize that both tasks have a strong correlation even when part of the cross-modal modeling does not have a direct visual support. Our transformer-encoder decoder represents a straightforward way to model both. In the final version we will follow a more standard naming of  the task, but still emphasize on the connection between them.

---

> ### Author Response · Authors · 2023-11-23
> **Usefulness in AV speaker diarization.**
>
> We must note that ASD and speaker diarization can be two independent tasks, for the case of audio only diarization. Certainly the AudioVIsual diarization and improved ASD pipeline task benefits the diarization taks. We see O2S as a method that can bridge the gap in both domains. It can directly estimate if the speech event must be attributed to visible speakers (thus an AV diarization method could be used for improved performance) or if the event must be attributed to unseen speech (thus audio only diarization must be used).

---

> ### Author Response · Authors · 2023-11-23
> **Writing Clarity**
>
> We thank the reviewer for the careful reading of our paper, we will fix the typos and reference errors in the final version.

---

> ### Author Response · Authors · 2023-11-23
> **Additional Questions**
>
> We can not tell how many potential out-of-screen speakers there are in a given scene in Ego4D. To the best of our knowledge no method or dataset provides such insight. Without this data we can not provide the requested ablation or at least an insight in the requested direction.
> However, we consider it is completely realistic to have multiple off-screen speakers, simply put the Field of View of most capture devices is far less than 360 degrees, even some people inside the FOV could be occluded, for example behind a third person or a grocery store stand.
> We are not aware of an official release of weights from LoCoNet for the Ego4D dataset. As outlined in the paper we train from weight initialized in Imagenet (2.5D) and Kinetics (3D)

---

### Official Review · Reviewer_dygL · 2023-10-31

**Soundness:** 3 good
**Presentation:** 4 excellent
**Contribution:** 2 fair
**Rating:** 6
**Confidence:** 3

**Summary:**

This paper proposes O2S, a framework for egocentric active speaker detection. Most active speaker detection literatures fall in commercial movies and social media videos, while egocentric videos are less investigated. O2S consists of three stages: 1) An audio encoder and a video encoder are employed to obtain audio features and visual face features. 2) A transformer serves as multimodal mixer to aggregate information from audio and video. 3) Another transformer serves as speech decoder to predict speech event for each face feature, audio feature, and an additional feature for the invisible camerawearer. There are some additional changes made for egocentric videos. First, face positions are added in the visual feature. Second, as egocentric videos may present many blurred faces due to fast motion, noisy faces are less contributed to the loss. Experiments are conducted on the Ego4D dataset for two tasks: Active Speaker Detection of visible targets
(vASD) and egocentric Active Speaker Detection (eASD).

**Strengths:**

1. The proposed O2S achieves the state-of-the-art performance on the Ego4D for both vASD and eASD.
2. The proposed method is a reasonable solution for egocentric active speaker detection.
3. The paper presentation and writing are very clear.

**Weaknesses:**

1. The authors should highlight the main differences between the proposed method and the previous 3rd person view active speaker detection methods. This is important to show the contribution of this proposed method.
2. I think Visual Token Representation and Weighted Visual Loss are the two unique contributions for the egocentric scenario. However, these two contributions are not significant. This brings back to the first concern: the authors should highlight the main differences compared to previous works especially on the main architecture.
3. Although the whole pipeline is reasonable, it is complicated. Does it need to first use a face detector to detect faces? In the main architecture of O2S, there are CNNs for encoding video and audio, and then there are transformers for mixing video and audio and decoding them. Why not encode and mix video and audio in just a single transformer? In my view, all the three stages can be simplified in one single transformer in principle.

**Questions:**

"In other words, the active speaker is more likely to appear near the center of the frame" I agree in most cases the active speaker appear around the center. But in some cases, the camerawearer may not look at the speaker or only turn eyeballs to look at the speaker. It may be more accurate to incorporate eye gaze location in the Visual Token Representation. After all, the XR device should already detected eye gaze.

---

> ### Author Response · Authors · 2023-11-23
> **Full Transformer Architecture**
>
> We politely direct the reviewert to the general discussion regarding the mAP metric to clarify the use of face detections in the validation set.
>
> We also clarify that, while we welcome the reviewer's suggestion, we must also bring into consideration that we can not simply  plug a multimodal transformer into the proposed head, we explain in detail:
>
> First, Since vASD predictions are made over individual face-crops. We still have to figure out a way to model features for each individual face-crop. Since the number of faces per frame can vary, we must resort either to simultaneous transformer forward passes (as is the case in O2S, but with CNN architectures), or a series of special tokens  delimiting the different faces and the individual face crops on each.
>
> Second,  it must be empirically determined if these facial features should be mixed prior to the encoder-decoder head proposed in O2S or earlier in the visual stream. This constitutes an additional hyper-parameter that should be explored. If it is determined that no attention is required between the individual face crops, we regress to the same scenario of O2S where  the CNN is replaced  with a visual transformer.
>
> Third, as we integrate the location data  of the face crops, an even more complex input stream for the visual transformer must be built. One might think about simply inputting the entire frame and encoding some ROIs corresponding to the faces. This will clearly incrates the memory allocation for the proposed network, and might be impractical as a portion of the visual objects on the frame have no direct relation with the task. For example the visual feature of a supermarket will not be truly useful in identifying the actual source of a speech event.
>
> Fourth, some of the structure proposed in the encoder-decoder transformer must remain to generate predictions which are temporally consistent and directly correspond to the input faces, in other words we can not simply mix all the data without any strategy to generate individual predictions over the inputs. Part of the complexity introduced in O2S is due to the requirement of directly supervising each token in the output with its corresponding ground-truth, if these correspondence is lost, the network supervision becomes significantly more challenging
>
> Certainly we share the same opinion of the reviewer regarding the versatility of transformers architectures for ASD tasks, but we also consider that it also brings extra challenges in a problem where visual and audio streams must be carefully aligned. We think this suggestion aligns better with the future work.

---

> ### Author Response · Authors · 2023-11-23
> **Novelty and Contribution**
>
> We politely invite the reviewer to check the general general comment posted in this regard

---

> ### Author Response · Authors · 2023-11-23
> **Additional Questions**
>
> We fully agree that the camera Field of View is not a direct estimation of the person's gaze, but rather a correlated variable. As outlined in the paper this empirical improvement of explicitly locating the face crops has already been explored  in datasets composed of  commercial movies.
> We see this additional information as common prior in the ASD task, regardless of the setup, egocentric or commercial movies. In the egocentric domain, we are the first to show the usefulness of this spatial prior.

---

### Official Review · Reviewer_kwFT · 2023-11-01

**Soundness:** 3 good
**Presentation:** 4 excellent
**Contribution:** 2 fair
**Rating:** 6
**Confidence:** 4

**Summary:**

Paper proposed a novel approach for the Active Speaker Detection (ASD) problem in egocentric data (esp. First Person Video (FSD)). This problem is relatively unexplored. The challenges for ASD in FPV is mainly to the "invisibility" of the camera wearer in the video which the SOTA ASD algorithms cannot handle correctly.

The proposed method uses multimodality to overcome this challenge via 3 building blocks: (i) Modality Encoder; (ii) Mutlimodal Mixer; (iii) Speech Decoder.

Experiments were performed to compare against SOTA ASD in FPV methods for the Ego4D dataset.

**Strengths:**

1. Paper's position that ASD in FPV is less research and proposed a novel method to overcome the specific issue for this problem is well explained and motivated.

2. Proposed method is somewhat novel and logical.

3. Experimental results are quite strong.

**Weaknesses:**

1. Problem statement is somewhat niche.
2. Novelty of proposed solution is limited as it's a special case of multimodality matching. The unseen visual features are replaced with a special token (c).

**Questions:**

No question.

**Details Of Ethics Concerns:**

Not applicable.

---

> ### Author Response · Authors · 2023-11-23
> **Niche Area of Study**
>
> We politely disagree with the reviewer's assertion, and instead bring his/her attention to the fact that there are multiple ASD papers with over 100 citations published in some of the important venues such as CVPR, ICCV and ECCV. Moreover, we must take into account that this area lacked a large-scale benchmark before the publication of the AVA-Active speaker dataset in 2019, and the Ego4D benchmark was only released last year. Overall this is an emerging line of work into the domain of multi-modal representation learning.

---

> ### Author Response · Authors · 2023-11-23
> **Novelty**
>
> We politely invite the reviewer to check the general general comment posted in this regard

---

### Author Response · Authors · 2023-11-22
**mAP Metric**

In addition to the individual replies to each reviewer, we would like to clarify our use of the mAP metric in the paper as a response of general interest. To the authors’ best understanding, the evaluation metric for the Ego4D validation set is effectively mAP, even though mAP@0.5 is written in the heading of Table 21 in the supplementary material of the Ego4D paper. We support our claim with the following points.

First, we kindly refer the reviewers and AC to the code released by Eric Zhongcong Xu (Ego4D author in the ASD task) which can be found in [R1]. There should be no confusion with the other Ego4D repository for the audiovisual task located at [R2], since that second repository simply redirects to Xu’s implementation of the TalkNet Ego4D baseline, the redirection can be found in [R3].

[R1] https://github.com/zcxu-eric/Ego4d_TalkNet_ASD

[R2] https://github.com/EGO4D/audio-visual

[R3] https://github.com/EGO4D/audio-visual/blob/main/active-speaker-detection/active_speaker/TalkNet_ASD/check_out_talknet.sh

Xu’s code repository shows that inference in the validation set is performed using the face crops provided by the ego4d annotation file, we refer the reviewers and AC to the code snippets at [R4] and [R5]. We also refer the reviewers to the issue #5 [R6] in the same repository, which also clarifies that the results are to be reproduced with the ground-truth face crops.

[R4] https://github.com/zcxu-eric/Ego4d_TalkNet_ASD/blob/ab9f345efc49fd70ed163c6cca674c3aff88e2b6/dataLoader.py#L230

[R5] https://github.com/zcxu-eric/Ego4d_TalkNet_ASD/blob/ab9f345efc49fd70ed163c6cca674c3aff88e2b6/dataLoader.py#L68

[R6] https://github.com/zcxu-eric/Ego4d_TalkNet_ASD/issues/5

Second, in the same thread, it is also asserted that the ground-truth face-crops can be used for inference in validation. And the tracked face crops are reserved for the test set (available only for the challenge). This means that only the results on the test-set for the Ego4D challenge are built from a tracker output. In such a scenario, the metric mAP@0.X makes sense, as the spatial accuracy of the detection must be taken into account.

Third, in the original versions of the two arxiv reports SPELL [R4]  and STHG [R5] that Reviewer 1nZN uses to support his/her claims, “mAP” is the metric used to report the performance of SPELL and STHG (Tables 3 and Table 1 respectively).  We further note that the TalkNet results of Ego4D are also included in the mentioned tables using “mAP”, just like in our main result table.

[R4] https://arxiv.org/pdf/2210.07764v1.pdf
[R5] https://arxiv.org/pdf/2306.10608v1.pdf

In a similar fashion, the paper by Wang et al. “LoCoNet” (whose authors overlap with the authors of Ego4D) also uses the mAP evaluation metric in the validation set. Once again the performance of Ego4D talk-net baseline is reported as mAP in the very same table.

To complement our shared remarks, we would like to raise the attention of the reviewers and AC to the fact that SPELL [R6] and STHG [R7] ArXiv reports were modified just before the release of the reviews (SPELL on October 29 and STHG on October 31). These new revisions introduced minor changes, but critically included the dual evaluation metric (mAP and mAP@0.5) in tables 3 and 1, with a much higher mAP (up to 12% more) than the original version.

[R6] Kyle Min, Intel Labs at Ego4D Challenge 2022: A Better Baseline for Audio-Visual Diarization, arxiv

[R7] Kyle Min, STHG: Spatial-Temporal Heterogeneous Graph Learning for Advanced Audio-Visual Diarization

While we welcome (and share) reviewers’ 1nZN enthusiasm to find the most recent literature in the ASD subject, to the point where he/she managed to spot two new ArXiv revisions submitted just 72 and 24 hours before the end of ICLR review period. We must be emphatic on the fact that the reviewers' claim for lack of state-of-the-art performance are substantiated by revisions made on the 28 of October and 31 of October. How could the authors be aware, compare, or discuss such results if their paper was submitted 1 month before (September 28) any of these ArXiv revisions were uploaded?.

We have searched for code releases of SPELL (Min et al.) and STHG (Min). An official release by the same author named “GraVi-T” [R8] was also updated on October 29, However, we could not make a more in-depth exploration, since the commit of that day did not provide a functioning code base to train, or at least to perform inference on Ego4D. It only contained a copy of the results in the newest version of [R6], [R7] and reflected on Reviewer 1nZN’s comments.
[R8] https://github.com/IntelLabs/GraVi-T

We conclude by re-stating that, to the best of our knowledge, the metrics and evaluation procedure are a direct and fair comparison to those in Ego4D validation set. We are eager to compare and include related works, but it is beyond the authors’ capability to benchmark and discuss results which are updated in ArXiv 1 month after the paper submission.

---

### Author Response · Authors · 2023-11-22
**Comparison against ArXiv Submisions**

In addition to supporting the selected metric, we kindly remind Reviewers (in particular 1nZN) that papers published within 4 months before the submission are considered concurrent work according to the conference rules [R5]. Therefore no paper should be penalized for not comparing with them. This is the case of the ArXiv report STHG originally released on 18 June of 2023.

[R5] https://iclr.cc/Conferences/2024/ReviewerGuide

Moreover, since both SPELL [R6] and STHG [R7] are 3-page non-peer review reports submitted to ArXiv, the conference rules also state that we are not required to compare with them, and definitely not to be penalized for the performance comparison.

[R6] Kyle Min, Intel Labs at Ego4D Challenge 2022: A Better Baseline for Audio-Visual Diarization, arxiv

[R7] Kyle Min, STHG: Spatial-Temporal Heterogeneous Graph Learning for Advanced Audio-Visual Diarization

---

### Author Response · Authors · 2023-11-23
**Novelty and Contribution**

We emphasize that the main difference in O2S is the joint prediction of the egocentric active speaker detection (egoASD) and the standar active speaker prediction. EgoASD is a unique element that emerges in the egocentric domain. As outlined in the paper, previous ASD approaches regard out-of-screen speech events as noise and work towards inhibiting the network response in these scenarios. We are the the first method to fully make sense of such events integrating the seamlessly into the egocentric ASD framework

To the best of our knowledge, all current methods try to model this prediction (egoASD) exclusively from the audio stream. O2S is the only method that can jointly model both prediction tasks, while showing a significant improvement in both metrics.

Given its simplicity (audio encoder, video encoder, and transformer head) we see O2S as a starting point to direct the attention of the community towards the detection of speech events in the egocentric domain.

---

### Meta-Review · Area_Chair_9swE · 2023-12-12

**Metareview:**

This paper received mixed but overall negative ratings (1, 3, 6, 6). Two reviewers who were in favor (`kwFT`, `dygL`) did not provide detailed explanations for why the paper should be accepted, while the other two (`2WYw`, `1nZN`) provided much more detailed and convincing arguments for rejection. There were several concerns about novelty, weak empirical evidence due to marginal improvements, and some doubts about experimental settings and evaluation metrics. The rebuttal addressed some of the concerns, but during the author discussion period, one critical issue stood out: the evaluation metric. The authors claimed that mAP should be the correct metric, while one of the reviewers asserted that mAP@0.5 should be used instead. The reviewer provided detailed justifications for why this is the case, citing the evaluation script provided by the official Ego4D challenge organizers. The authors did the same, citing a repository from one of the authors of the Ego4D paper. To verify the correctness of the claim, this meta-reviewer has contacted the Ego4D ASD challenge organizers, and they confirmed that mAP@0.5 is indeed the correct metric for the official challenge since a face detector is involved. Given the inaccuracies involved in the paper, we are recommending rejection at this time in the hope that the authors will adopt the correct evaluation metric.

**Justification For Why Not Higher Score:**

There is an important issue with the evaluation metric.

**Justification For Why Not Lower Score:**

N/A

---

### Decision · Program_Chairs · 2024-01-16

Reject